# Microbial Synthesis of Heme *b*: Biosynthetic Pathways, Current Strategies, Detection, and Future Prospects

**DOI:** 10.3390/molecules28083633

**Published:** 2023-04-21

**Authors:** Qiuyu Yang, Juntao Zhao, Yangyang Zheng, Tao Chen, Zhiwen Wang

**Affiliations:** 1Frontier Science Center for Synthetic Biology and Key Laboratory of Systems Bioengineering (Ministry of Education), Tianjin University, Tianjin 300072, China; 2SynBio Research Platform, Collaborative Innovation Center of Chemical Science and Engineering (Tianjin), School of Chemical Engineering and Technology, Tianjin University, Tianjin 300072, China

**Keywords:** heme *b*, 5-aminolevulinic acid, biosynthetic pathways, metabolic engineering strategies, detection

## Abstract

Heme *b*, which is characterized by a ferrous ion and a porphyrin macrocycle, acts as a prosthetic group for many enzymes and contributes to various physiological processes. Consequently, it has wide applications in medicine, food, chemical production, and other burgeoning fields. Due to the shortcomings of chemical syntheses and bio-extraction techniques, alternative biotechnological methods have drawn increasing attention. In this review, we provide the first systematic summary of the progress in the microbial synthesis of heme *b*. Three different pathways are described in detail, and the metabolic engineering strategies for the biosynthesis of heme *b* via the protoporphyrin-dependent and coproporphyrin-dependent pathways are highlighted. The UV spectrophotometric detection of heme *b* is gradually being replaced by newly developed detection methods, such as HPLC and biosensors, and for the first time, this review summarizes the methods used in recent years. Finally, we discuss the future prospects, with an emphasis on the potential strategies for improving the biosynthesis of heme *b* and understanding the regulatory mechanisms for building efficient microbial cell factories.

## 1. Introduction

Hemes belong to a small subgroup of the tetrapyrrole family, which is characterized by the combination of a ferrous ion and a porphyrin macrocycle. The ‘true’ hemes possess a fully oxidized porphyrin macrocycle, including heme *a*, heme *b*, heme *c*, and heme *o* [1]. Hemes are an important class of prosthetic molecules that play roles in a number of biological processes. The most widespread and ubiquitous is heme *b*, which plays an important role in transporting oxygen as part of hemoglobin. In addition, heme *b* is a cofactor for many enzymes, such as myoglobin, cytochrome P450, and peroxidases, and plays significant roles in catalysis, transcription, signaling, and electron transfer [2]. Due to its unique physicochemical properties, heme *b* has a wide range of commercial applications in the pharmaceutical, food, and fine chemical industries [3,4,5]. In addition, recent studies have also shown the potential uses of heme *b* in biochemical analysis and biocatalysts [6].

Currently, heme *b* is obtained primarily through chemical synthesis or its isolation from plant tissues and animal blood using organic extraction or enzymatic hydrolysis [7,8]. However, these approaches are complex and environmentally unfriendly, e.g., acetone, the main raw material used for the heme *b* extraction, is not only volatile but also physiologically toxic [8,9]. Alternative microbial production routes for heme *b* have been intensively studied and are well suited for solving the shortcomings of these methods, leading to large-scale, low-cost production. In 2003, Kwon et al. first assembled the entire heme *b* synthetic pathway in *Escherichia coli* using a three-plasmid system, resulting in a porphyrin titer of 90 μmol/L; however, the heme *b* titer was only 3.3 ± 0.3 μmol/L [10]. In recent years, three biosynthetic pathways of heme *b* have been elucidated [11]. Zhao et al. constructed the *E. coli* strain HAEM7 by optimizing the expression level of the heme *b* synthesis genes, deleting the competing pathways, and overexpressing a heme export protein [9]. After further optimization of the fermentation conditions, the strain achieved the highest total heme *b* titer of 1034.3 mg/L [12]. Additionally, detection methods based on UV spectrophotometry are gradually being replaced by new technologies based on fluorescence detection, high-performance liquid chromatography (HPLC), and biosensors [12,13,14]. Increasingly diverse and new heme *b* detection methods are being established, which greatly promote the development of the microbial synthesis of heme *b*.

This review first systematically introduces the three main heme *b* biosynthetic pathways, which are used as the basis for the first comprehensive summary of the current development strategies for promoting the microbial biosynthesis of heme *b*. The establishment of the heme *b* detection methods in recent years is also reviewed for the first time. Finally, we look ahead and propose new engineering strategies for constructing high-performance microbial cell factories for the heme *b* synthesis, providing insight for industrial production.

## 2. Biosynthetic Pathway of Heme *b*

The capacity to synthesize heme *b* is very common but not ubiquitous, and there are organisms with a complete absence of a tetrapyrrole biosynthetic capacity or incomplete pathways. The available research suggests that the overall biosynthetic pathway of heme *b* can be divided into three parts (Figure 1): (i) the synthesis of the precursor 5-aminolevulinate (5-ALA) via the C4 or C5 pathway, (ii) the synthesis of the intermediate metabolite uroporphyrinogen III (UPG III) via a conserved core pathway, and (iii) the synthesis of heme *b* via the protoporphyrin-dependent (PPD) pathway, coproporphyrin-dependent (CPD) pathway, or siroheme-dependent (SHD) pathway.

### 2.1. Biosynthetic Pathways of the Precursor 5-ALA

The common precursor of tetrapyrroles, 5-ALA, is synthesized via two completely different biosynthetic pathways. The C4 pathway (Shemin pathway) was the first to be discovered, using succinyl-CoA and glycine as the substrates (Figure 1). It is mainly found in metazoans, fungi, and alphaproteobacteria [11]. For a long time, this was thought to be the only pathway for 5-ALA biosynthesis. However, in 1973, Beale et al. confirmed the existence of an alternative 5-ALA biosynthesis pathway, called the C5 pathway (Beale pathway), through their studies on cucumber cotyledons [15]. This route, which uses glutamate as the initial substrate, is predominantly found in plants, archaea, and most bacteria [11]. Generally, only one of these pathways is present in each organism, but a small number of organisms contain both pathways, such as *Euglena gracilis* [16].

The C4 pathway requires only one step to synthesize 5-ALA, which is catalyzed to the pyridoxal-5′-phosphate (PLP)-dependent ALA synthase (AlaS). This is a homodimeric protein, containing an active-site lysine that is covalently bound to the PLP cofactor, which catalyzes the condensation of succinyl-CoA and glycine, releasing carbon dioxide and coenzyme A in the process [16]. As shown in Figure 1, the C5 pathway is more complex than the C4 pathway, requiring three steps to synthesize 5-ALA from glutamate. First, glutamate is converted into Glu-tRNA in the presence of glutamyl-tRNA synthetase (GluTS). The next step is catalyzed using the NADPH-dependent glutamyl-tRNA reductase (GluTR). Its highly conserved cysteine residues bind to glutamyl-tRNA and release Glu^tRNA^ to form a thioester intermediate, which is then reduced to glutamate-1-semialdehyde (GSA) [17]. Finally, glutamate-1-semialdehyde-2,1-aminomutase (GsaM) catalyzes an intramolecular amine transfer, resulting in the conversion of GSA to 5-ALA [18]. GsaM also requires PLP as a cofactor and has a structural similarity to AlaS. Since the GSA intermediate is unstable, GluTR and GsaM usually tend to form the stable complexes for the substrate channeling [19].

### 2.2. Formation of the Common Tetrapyrrole Core UPG III

UPG III, the common core of tetrapyrroles, is formed through the condensation of eight 5-ALA molecules in a highly conserved three-enzyme pathway (Figure 2). These three enzymes can be considered as the backbone of the biological tetrapyrrole synthesis tree, with a variety of branches formed by modifications and different metals [5].

At the beginning, the asymmetric condensation of two 5-ALA molecules catalyzed by porphobilinogen synthase (PbgS) forms the monopyrrole porphobilinogen (PBG) [20]. This enzyme contains two 5-ALA binding sites, called A and P. The first 5-ALA molecule that binds to the enzyme forms the propionate side chain of PBG, while the second forms the acetate side chain [21]. The enzyme requires metals to be active in both the eukaryotes and prokaryotes [22], and the metal requirement determines the features of the A site. In the next step, the pyrrole building block PBG is used for the large-loop assembly. Hydroxymethylbilane synthase (HmbS) catalyzes the sequential deamination of the aminomethyl groups of the four PBG molecules, and the deaminated pyrroles are strung together to produce the linear tetrapyrrole hydroxymethylbilane (HMB) [23]. For the HmbS catalysis, two PBG molecules are required to form a unique dipyrromethane cofactor through covalent linkage, and the enzyme-bound tetrapyrrole is formed after the sequential addition of PBG, followed by the cleavage of the reaction product HMB, leaving the holo-enzyme behind [24]. Finally, uroporphyrinogen synthase (UroS) cyclizes HMB and notably reverses the D-ring, producing the only asymmetric isomer, UPG III [25]. Although a simple cyclization can produce a type I isomer, the ring reversal was favored by evolution [26]. UroS is highly sensitive to proteolysis and heat denaturation, resulting in a significantly lower intracellular abundance than HmbS [11].

### 2.3. Multiple Pathways for Synthesizing Heme b

Three heme *b* synthesis pathways have been discovered in living organisms (Figure 3). The PPD and CPD pathways are known as the two branches of the classical heme *b* synthesis pathway. The PPD pathway is found mainly in eukaryotes and many gram-negative bacteria, while the CPD pathway is found mainly in gram-positive bacteria. As shown in Figure 2, the decarboxylation of UPG III catalyzed by uroporphyrinogen III decarboxylase (UroD) [27] forms coproporphyrinogen III (CPG III), which is the last step before the branching pathways. In both classical pathways, the four methyl groups of heme *b* are produced by the stepwise decarboxylation of the acetate side chain of UPG III, with four methyl groups produced at the C2, C7, C12, and C18 positions [28]. Hence, no methylation of the tetrapyrrole backbone occurs in the classical pathways. The alternative siroheme-dependent (SHD) pathway is considered to be an evolutionary remnant from an anaerobic world [29]. Some archaea and sulfate-reducing bacteria rely on this pathway to synthesize siroheme and then convert it into heme *b* [30]. Unlike the two classical routes, the SHD pathway involves the S-adenosyl-L-methionine (SAM)-dependent methylation of the C2 and C7 positions of the tetrapyrrole backbone [31].

#### 2.3.1. The Protoporphyrin-Dependent (PPD) Branch

In the PPD branch (Figure 3a), the two propionic acid side chains on the A- and B-rings of CPG III are oxidized and decarboxylated to their corresponding vinyl groups to form protoporphyrinogen IX (PPG IX) [32]. Two main types of enzymes catalyze this reaction. One is the oxygen-dependent coproporphyrinogen III decarboxylase (CgdC), which is present in most eukaryotes and a small percentage of gram-negative bacteria [33]. The other is the oxygen-independent coproporphyrinogen III dehydrogenase (CgdH), an exclusively bacterial enzyme and a member of the free radical SAM protein family [34]. They are completely unrelated in their structure and mechanism, but both first catalyze the modification of the A-ring and form a monovinyl monopropionate deuteroporphyrin intermediate [35].

Next, PPG IX undergoes a six-electron oxidation to produce protoporphyrin IX (PP IX). Three completely different enzymes have been found to catalyze this reaction, including the oxygen-dependent protoporphyrinogen IX oxidase (PgoX), and two oxygen-independent protoporphyrinogen IX dehydrogenases (PgdH1 and PgdH2). The FAD-containing PgoX shares a certain degree of sequence similarity with coproporphyrinogen oxidase (CgoX) from the CPD pathway, which is found in eukaryotes and a few gram-negative bacteria [36]. PgdH1 is a member of the long-chain flavodoxin family and is found mainly in gammaproteobacteria, such as *E. coli.* It couples the oxidation of PPG IX to the anaerobic respiratory chain rather than using oxygen directly [37]. PgdH2 is a membrane-bound protein, and similar to PgdH1, it does not directly interact with the electron acceptors, such as oxygen [38]. However, little is known about PgdH2, although it is present in almost two-thirds of gram-negative bacteria [39].

The final step is the insertion of a ferrous ion into PP IX, catalyzed by protoporphyrin IX ferrochelatase (PpfC). This enzyme is also poorly understood, but there is evidence that a [2Fe-2S] cluster may be a widespread feature of ferrochelatases [11,40]. However, its exact role is unclear. It is worth considering the source of the ferrous irons required by ferrochelatase due to the activity of the iron metabolism. In the organisms that were studied, there was no stoichiometric link between the iron reduction and the heme synthesis, although it was tightly linked to the cellular respiratory chain [41]. The complex and long pathways, such as the heme *b* synthesis, seem to employ multi-enzyme complexes in most organisms. Studies on eukaryotes confirmed the existence of multi-enzyme complexes containing the heme *b* synthetases [42]. However, there is still a lack of sufficient data to prove the existence of the same complexes in bacteria, and the only available data are those involving the PgoX and PpfC complexes [43].

#### 2.3.2. The Coproporphyrin-Dependent (CPD) Branch

In evolutionary terms, the CPD pathway can be considered a transitional form between the SHD and PPD pathways [44]. It shares the same precursor, CPG III, with the PPD pathway, and the methyl group of the tetrapyrrole backbone is not involved in the reaction. Similar to the SHD pathway, it does not have protoporphyrinogen or protoporphyrin as intermediates (Figure 3b).

In the CPD pathway, the six electrons of CPG III are first oxidized to form coproporphyrin III (CP III). This implies that the macrocyclic oxidation in the CPD pathway occurs earlier than in the PPD pathway [11]. This conversion from a flexible, cyclic tetrapyrrole porphyrinogen to a fully conjugated, planar macrocyclic porphyrin is achieved through porphyrin oxidation using coproporphyrinogen oxidase (CgoX). CgoX is a soluble monomeric protein containing flavin adenine dinucleotide (FAD), which has two distinct structural differences compared to PgoX. In CgoX, the putative active-site pocket is larger and has more positively charged residues [36]. Based on these two features, CgoX has a higher affinity for CPG III.

The insertion of the ferrous ions to form Fe-coproheme is the next step in the CPD pathway. The iron chelatase involved in this step is coproporphyrin ferrochelatase (CpfC), which is a soluble monomeric protein [45]. It has an obvious structural homology with PpfC, but it can use CP III as a substrate, while PpfC cannot. This may be due to a lid consisting of a dozen residues on one side of the active site of PpfC, which can close the binding pocket during the catalytic cycle [46,47]. In this closed position, there is not enough space in the pocket to hold CP III. CpfC does not have a lid and can, therefore, accept CP III as a substrate, but this also means that its active site remains open during catalysis.

The last reaction, catalyzed using Fe-coproheme decarboxylase (ChdC), is the oxidative decarboxylation of the two propionic acid side chains on the coproheme pyrrole rings A and B to produce the corresponding vinyl groups [48]. The decarboxylation reaction catalyzed using ChdC requires the presence of a proton acceptor.

Under aerobic conditions, it can utilize the H_2_O_2_ produced by the earlier pathway enzyme CgoX [38], which is not feasible under anaerobic conditions. A gene similar to *chdH*, *ahbD*, was identified in the genomes of many gram-positive bacteria, likely encoding a free radical SAM enzyme that perhaps also catalyzes this reaction [49]. This enzyme will be described in detail in the section on the final step of the SHD pathway.

#### 2.3.3. The Siroheme-Dependent (SHD) Pathway

The SHD pathway was only fully deciphered in 2011, confirming that siroheme is the key intermediate (Figure 3c) [30]. Siroheme is generated by a pathway consisting of three enzymes, including an SAM-dependent uroporphyrinogen III methyltransferase (SumT), an NAD^+^-dependent precorrin-2 dehydrogenase (PcdH), and a sirohydrochlorin ferrochelatase (ShfC) [50]. In this process, the C2 and C7 positions of UPG III are methylated to produce precorrin-2, which is then dehydrogenated to produce sirohydrochlorin, which is finally converted into siroheme through the iron insertion. In some organisms, such as *E. coli*, a multifunctional protein, CysG, exists, which contains the enzymatic activities of the above three enzymes and can directly convert UPG III into siroheme [51].

The latter three steps of the pathway are encoded using a series of *ahb* genes. Firstly, in the presence of AhbA and AhbB, the acetic acid side chains attached to C12 and C18 are decarboxylated to generate 12,18-didecarboxysiroheme (DDSH). The siroheme decarboxylase is either a heterodimer of the two subunits AhbA and AhbB, or a genetically encoded fusion of AhbA and AhbB. AhbC oxidizes and removes the C2 and C7 acetic acid side chains of DDSH to produce Fe-coproheme. Finally, AhbD decarboxylates the propionic acid side chains attached to C3 and C8 to produce heme *b* [30]. AhbC and AhbD both belong to the radical SAM enzyme family and, therefore, both contain an [4Fe-4S] cluster [52].

## 3. Metabolic Engineering Strategies for Heme *b* Biosynthesis

A series of metabolic engineering strategies were developed to achieve the biotechnological production for heme *b* (Figure 4). Screening and comparing the heme *b* biosynthetic pathways provided a good basis for the construction of the heme *b* microbial cell factories. Increasing the supply of the key precursor 5-ALA and balancing the expression levels of the gene encoding enzymes in the pathway from 5-ALA to heme *b* accelerated the development of biosynthetic heme *b*. In addition, the accumulation of heme *b* was further promoted by blocking the downstream pathways. Since excess intracellular heme *b* can be toxic to the host, improving the efficiency of the cellular export facilitated a balance between the growth and production. Furthermore, by modifying the intracellular iron ion metabolism, the iron concentration could be optimized to achieve an efficient heme *b* synthesis. The recent advances in the research on the heme *b* biosynthesis are summarized in Table 1.

### 3.1. Screening and Comparing the Heme b Biosynthetic Pathways

In addition to the more conserved four-pyrrole core pathway from 5-ALA to UPG III, two 5-ALA synthesis pathways and three heme *b* synthesis pathways provided more options for the construction of the microbial cell factories. In order to efficiently produce heme *b*, the abilities of the two pathways for producing 5-ALA were compared [53]. In *E. coli*, the *alaS_RS_*, *coaA*, and *maeB* genes were co-expressed to establish the C4 pathway. The *gluTS*, *gluTR*, and *gsaM* genes were co-expressed to build the C5 pathway. After a flask cultivation in the Luria-Bertani medium, the C5 strain produced 1.74 ± 0.08 g/L of 5-ALA, over 5-fold more than the C4 strain. However, due to the unique features of each chassis strain, the experimental results may vary greatly in different host organisms. In *Corynebacterium glutamicum*, plasmids were used to individually overexpress the C4 and C5 pathways to compare their ability to produce the precursors and heme *b*. The study showed that the strains expressing the C4 pathway produced more 5-ALA, but the titer of heme *b* produced via the C5 pathway was 67% higher than via the C4 pathway [54].

Among the three known pathways of the heme *b* biosynthesis, only the PPD and CPD pathways have been applied for the microbial synthesis of heme *b* (Figure 4a). Seok et al. [55] compared the CPD and PPD pathways for the synthesis of heme *b* in *C. glutamicum* ATCC 13826. According to the thermodynamic data (Appendix A), the ΔG^0,^ value of the reaction catalyzed using ferrochelatase was 244.65 kJ/mol, indicating that this might have been a bottleneck. The authors, therefore, used a dual plasmid system, one plasmid overexpressing the *gluTR* and *gsaM* genes of the C5 pathway to ensure the supply of 5-ALA, and the other overexpressing *cgdC* and *chdC* to encode the key enzymes of the two pathways, respectively. The overexpression of the *chdC* gene in the CPD pathway resulted in a heme *b* production of 17.66 ± 0.43 mg/L, which was 50% higher than the *cgdC* gene in the PPD pathway. This was consistent with the thermodynamic prediction that the ΔG^0^, value of the CPD pathway was 447.22 kJ/mol lower than that of the PPD pathway (Appendix A).

An in vitro thermodynamic analysis can be used to calculate the Gibbs free energy of each enzyme in the path and identify the hypothetical rate-limiting steps [56]. However, this approach has limitations in the face of the complex intracellular metabolic environment, and the choice of the best pathway still requires the careful consideration of multiple factors. In addition, it should be noted that the SHD pathway has not been used for the biotechnological production of heme *b*.

### 3.2. Increasing the Supply of the Key Precursor 5-ALA

Eight 5-ALA molecules are required to synthesize one molecule of heme *b*. Therefore, an adequate supply of 5-ALA is a prerequisite for the efficient synthesis of heme *b* in microorganisms. Metabolic engineering strategies for increasing 5-ALA biosynthesis have been summarized in many previous reports [57,58]. The TCA cycle and the 5-ALA synthesis pathway were the focuses of the metabolic engineering modifications aimed at enhancing the supply of the precursor 5-ALA for synthesizing heme *b* (Figure 4b).

#### 3.2.1. Engineering the Metabolic Flow in the TCA Cycle

Since 5-ALA is generated from glycine and succinyl-CoA, which in turn are derived from the metabolites of the TCA cycle, increasing the flux through the TCA cycle can effectively improve the supply of 5-ALA. Blocking the synthesis of the acetate and lactate catalyzed by phosphate acetyltransferase (encoded by *ptA*) and lactate dehydrogenase (encoded by *ldhA*) ensures that more carbon enters the TCA cycle [26,53]. The knockdown of these two genes led to a 10% increase in the heme *b* production in an *E. coli* strain that overexpressed the genes involved in the heme *b* synthesis [9]. To increase the availability of succinyl-CoA, a NADPH-dependent malic enzyme (MacB) and a C4-dicarboxylic acid transporter protein (DctA) were considered for the study of the heme *b* production. MacB catalyzed the oxidative decarboxylation of malate to pyruvate in the presence of divalent metal ions [59] and DctA facilitated the transport of succinate into the cells. When the *alaS_RS_* gene was overexpressed together with the *macB* and *dctA* genes in *E. coli*, 38.4 mg of intracellular heme *b* could be extracted from 6 L of the fermentation broth, while it was almost undetectable in the control strain [60].

In addition to succinate, coenzyme A is also essential for succinyl-CoA synthesis. Pantothenate kinase (CoaA) produces 4-phosphopantothenate using an activated phosphate group from ATP [61], which ultimately increases the CoA pool. The co-expression of the *coaA* gene from *E. coli* in combination with the *alaS*, *macB*, and *dctA* genes resulted in a CoA concentration of 75.7 nmol/g-DCW, which was 60% higher than the control strain. The heme *b* concentration (0.48 µmol/g-DCW) was 90% higher than the control. It is noteworthy that CoA can inhibit pantothenate kinase by competitively binding to the ATP binding site (Lys101) [62]. As PEP carboxykinase (PckA) increases the intracellular ATP levels under glycolytic conditions [63], the addition of this gene might increase the effect of CoaA by reducing the feedback inhibition.

#### 3.2.2. Optimizing the 5-ALA Biosynthetic Pathway

The C4 pathway was the first pathway to be applied in the study of the biosynthesis of heme *b*. The AlaS orthologs from *Rhodobacter capsulatus* (*alaS_RC_*) [10,54], *Rhodobacter sphaeroides* (*alaS_RS_*) [9,60,64] and *E. coli* (*alaS_EC_*) [65] were overexpressed to increase the heme *b* production. In addition, the metabolic characteristics of AlaS can be altered to facilitate the heme *b* accumulation though the modification of the key sites. AlaS^fbr^ was produced by inserting two lysine residues (KK) between the second and third residues of the native AlaS to prevent the unusual proteolysis-mediated feedback inhibition induced by excess heme *b* [66]. Finally, the genomically integrated multicopy expression of the *alaS* gene was also studied. The *saccharomyces cerevisiae* strains were constructed based on the random integration of the *alaS* gene under the control of the P_TEF1_ promoter at more than 100 δ locus sites in the genome of the CEN.PK2-1C host. Among ten randomly selected strains, the copy number of the integrated *alaS* genes ranged from two to seven and the highest heme *b* titer was 6.1-fold higher than that of the control group [67].

The three genes involved in the synthesis of heme *b* in the C5 pathway have been analyzed in detail [13]. The shuttle vector pEC-XK99E was used to overexpress each of the three genes in *C. glutamicum*, but only the strain overexpressing the *gluTR* gene showed a significant 4.6-fold increase in the heme *b* levels. This indicated that the generation of GSA is a rate-limiting step, which is consistent with the thermodynamic analysis, as it had an extremely positive ΔG^0^ value. Subsequently, the GluTS, GluTR, and GsaM enzymes from the different sources were expressed in combination, and five different recombinant strains were established. Among them, the strains co-expressing with the *gluTR* gene from *Salmonella arizonae* with the *gluTS* and *gsaM* genes from *E. coli* produced the highest heme *b* titer of 4.22 ± 0.62 mg/L. Since it was shown that overexpression of the *gluTS* gene may adversely affect the overexpression of the other genes and cell growth [13,68], most researchers emphasized only the expression of the GluTR and GsaM enzymes in the C5 pathway [53,54].

To some extent, the consolidation of either the C4 or C5 pathway allows for the accumulation of 5-ALA, the crucial precursor of heme *b*. To further increase the flux of the 5-ALA synthesis pathway, the C4 and C5 pathways were overexpressed simultaneously. The strain with both pathways produced 22.95 ± 0.63 mg/L of heme *b*, which was higher than in the single pathway strains and 4.49-fold higher than the wild-type strain [54]. However, the increase in the number of pathways also led to an escalation in the complexity of the metabolic regulatory network.

### 3.3. Balancing the Expression Levels of the Gene Encoding Enzymes in the Pathway from 5-ALA to Heme b

Metabolic engineering reconfigures the cell’s biochemical network by manipulating the endogenous genes or introducing the heterologous genes to direct the conversion of the renewable raw materials into value-added compounds. However, the readjustment of the native metabolism of microorganisms often leads to serious imbalances in the pathway fluxes, which can have detrimental effects. The accumulation of intermediates can lead to a feedback inhibition of the upstream enzymes and the formation of unwanted by-products. Furthermore, some toxic intermediates may interfere with the cell growth [69]. This requires researchers to look at the cellular metabolism on a holistic level in order to identify the best balance.

Currently, the optimization of the heme *b* pathway is mainly focused on multiple modular metabolic engineering (MMME), which relies on plasmids with different copy numbers (Figure 4c) [70]. Kwon et al. [10] made the first attempt to reconstruct the heme *b* biosynthesis pathway in *E. coli* using three compatible plasmids, including the C4 and PPD pathways. Each plasmid carried one or more heme *b* biosynthesis genes cloned from various bacteria, mostly from *E. coli*. All the heme-specific genes were expressed using a *lac* promoter. In detail, the *alaS*, *pbgS*, *hmbS*, and *uroS* genes were co-expressed from the pACmod plasmid, *uroD* and *cgdC* were co-expressed from the pBBR1-MCS2 plasmid, and the *pgoX* and *ppfC* genes were overexpressed from the pUCmod plasmid. In this study, the heme *b* concentration reached 3.3 ± 0.3 μmol/L. In addition, the combined optimization of the genes in the C5 and PPD pathways using three plasmids with different copy numbers increased the heme *b* production from 0.5 mg/L to 3.8 mg/L [9]. However, the unstable expression of the extrachromosomal DNA had a significant impact on the synthesis of heme *b*.

To further improve the heme *b* production with a precise regulation, more efficient target genes were screened. In the PPD pathway, the co-expression of the *pbgS* and *ppfC* genes using the medium copy number plasmid pETDuet-1 resulted in a significant accumulation of intermediates. On this basis, the five genes *hmbS*, *uroS*, *uroD*, *cgdC*, and *pgoX* were overexpressed separately, and all the strains except the strain overexpressing *pgoX* showed positive changes in the intermediate profiles compared to the control. However, no production of heme *b* was detected. By contrast, when the five genes were overexpressed simultaneously, the heme *b* titer reached 0.53 mg/L [9]. This might have been due to the feedback inhibition of PbgS by PPG IX [71]. The high expression of PgoX reduced the accumulation of PPG IX, which led to an increase in the production of heme *b*. Ko et al. [54] produced heme *b* through the CPD pathway and showed that the overexpression of the *chdC* gene was conducive to heme *b* and CP III production. However, the catalytic process of CgoX produced hydrogen peroxide, and the catalytic effect of CpfC was strictly regulated by the intracellular iron ions. As a consequence, the overexpression of CgoX and CpfC alone was not conducive to an efficient product synthesis. The final result was obtained using two plasmids, pEKEx2-*gluTR^M^*-*gsaM*-*alaS-dtxR* and pMTZ-*cgoX*-*cpfC*-*chdC*, and produces a total heme *b* titer of 28.66 ± 0.67 mg/L. In addition, the assembly of the rate-limiting enzymes by the DNA-guided scaffolds can be used to further optimize the heme *b* biosynthesis flux [72].

### 3.4. Blocking Downstream Pathways

Similar to the regulatory strategies used for the other products, the downregulation or knockdown of the endogenous heme *b* degradation pathway can favor the heme *b* accumulation. As shown in Figure 4d, two homologous enzymes, YfeX and EfeB, which are present in the cytoplasm and periplasm of *E. coli*, respectively, can recover iron from heme *b* without destroying the porphyrin macrocycle, suggesting that they may be dehydrogenases [73]. However, the dehydrogenation of heme *b* to free porphyrin was not detected in the purified YfeX or the cell extracts of the *E. coli* strains with a high expression of the *yfeX* gene. This indicated that the accumulation of protoporphyrin when YfeX was overexpressed in *E. coli* may be due to the intracellular oxidation of the endogenously synthesized protoporphyrinogen rather than the dissociation of the exogenously provided heme *b* [74]. At the same time, the overexpression of the *yfeX* gene affected the heme *b* homeostasis [75]. By removing the *yfeX* gene, the production of heme in *E. coli* increased to 6.2 mg/L, indicating that YfeX may be involved in the degradation of heme *b* or its intermediates [9].

In *C. glutamicum*, heme oxygenase (HmuO) catalyzes the degradation of heme *b* to biliverdin, CO, and free iron. Due to the complexity of the heme *b* regulatory network, the *hmuO* transcription is subject to both inhibition by DtxR and activation by HrrA. The latter is required for basal level expression of *hmuO*, but the DtxR repression imposes an additional threshold for transcriptional activation. DtxR can only dissociate from the promoter of the *hmuO* gene and allow HrrA to induce the full activation only when the intracellular iron pool is depleted [76]. However, when attempts were made to disrupt the *hmuO* gene by producing a nonsense mutation to completely prevent the heme degradation, the specific growth rate, and the heme *b* production of the engineered strain were significantly lower than the control strain [54]. This may be due to the toxic effect of the high heme *b* concentrations on the strain. Therefore, reducing its expression level rather than knocking it out completely may be the best strategy.

### 3.5. Improving the Efficiency of the Cellular Export

Heme *b* is an important compound in the aerobic electron transport chain, but paradoxically, the high redox potential of heme *b* makes it a liability, since it is toxic in high concentrations. Although, the underlying mechanisms are still poorly understood, non-iron metal porphyrins have also shown a significant antibacterial activity [77]. This indicates that certain porphyrin structures have a general iron-independent toxicity. Among the various mechanisms of the heme *b* tolerance, the active export of heme or its toxic metabolites to the extracellular space can improve the product tolerance (Figure 4e).

In *E. coli*, the cytochrome *c* export protein (CcmABC) may be related to the heme export factors [78]. The heme *b* secretion rate was increased to 63% in the fed-batch fermentation by overexpressing the *ccmABC* gene [9]. According to the genomic information of *C. glutamicum* ATCC 14067, the heme-regulated transporters A and B (HrtA and HrtB), as well as the cytochrome C export protein (CcsA) may be involved in the process of the heme *b* secretion outside of the cell. A comparison between two recombinant strains of *C. glutamicum* overexpressing the *hrtAB* and *ccsA* genes showed that HrtAB was more conducive for the heme *b* production, with an extracellular secretion rate of 9.25% [54]. Most gram-positive bacteria take up heme *b* as a source of iron through an ABC heme transport system consisting of heme-binding proteins and heme transport proteins. In *C. glutamicum*, HrrS, HtaA, and HmuT are the three typical heme-binding membrane proteins that play key roles in transporting heme into the cytoplasm and regulating the heme biosynthesis [79,80]. The induction of nonsense mutations in these three genes using the CRISPR-Cas12a system resulted in a total heme *b* yield of 35.65 mg/L [54].

In addition to the heme-binding proteins, the fatty acids bound to the cell wall also have a high affinity for heme *b* [81]. Mycolic acid is a major fatty acid in the outer membrane of *C. glutamicum*, and ethambutol (ETB) can inhibit its production, thus promoting the extracellular secretion of heme *b* [82]. A modified CGXII medium with ETB was used for the supplemented batch fermentation of the heme-producing engineered strain ΔSAT: ALSdtE-YHQBA, and the heme *b* secretion rate was eventually increased to 91.25% in a 2 L fermenter. However, this modification had a negative effect on the total heme *b* production and cell growth [54].

### 3.6. Optimizing the Iron Concentration by Engineering the Intracellular Iron Ion Metabolism

Iron is an essential limiting nutrient for microorganisms [83]. The iron-limited conditions not only inhibited the growth of *E. coli*, but also resulted in a 2.5-fold decrease in the intracellular heme *b* levels compared to the control [84]. However, when excess iron was added exogenously, it inhibited the overproduction of 5-ALA and porphyrins [85], which also adversely affected the heme *b* synthesis. Thus, a more stringent regulation of the intracellular iron ion metabolism is required to maintain the balance between the cell growth and the heme *b* production (Figure 4f).

In *E. coli* and many other bacteria, the ferric uptake regulator (Fur) mediates the global transcriptional control of iron [86]. When the *fur* genes were homologously overexpressed in *E. coli*, the intracellular iron ion content was reduced and the transcript levels of most of the heme *b* synthesis genes were increased to varying degrees, except for *gluTR*. This was due to the large amount of Fur protein that binds to Fe^2+^ to form the Fur-Fe^2+^ complex, inhibiting the expression of the iron ion transport system and promoting the expression of the gene encoding ferritin-related proteins. However, the overexpression of the *fur* gene alone did not increase the heme *b* production, likely due to an insufficient supply of 5-ALA [84]. In addition to direct regulation, Fur-Fe^2+^ is also able to achieve indirect regulation through a small regulatory RNA molecule, called RyhB. When RyhB was constitutively expressed in the engineered *E. coli*, the 5-ALA production increased by 16%, but the heme *b* content was significantly reduced. This may be due to RyhB interacting with *pbgS* and *ppfC* by binding to specific sites in the DNA, thereby repressing the transcription of these genes [87].

Gram-positive bacteria with a high genomic GC content use the diphtheria toxin repressor (DtxR) protein as an iron regulatory system instead of Fur. The DtxR proteins operate in a similar manner to the Fur proteins, although they are not phylogenetically related [88]. The mutants lacking the *dtxR* gene can only be grown in a low-iron medium [89]. When the *dtxR* gene was further overexpressed in addition to the overexpression of *alaS*, *gluTR*, *gsaM*, *cgoX*, *cpfC*, and *chdC*, there was a 3.17-fold increase in the heme *b* production. This may be because the DtxR protein alleviated the inhibitory effect of the iron overload on heme *b*. At the same time, the transcriptional analysis showed that the expression of most genes related to the heme *b* synthesis and those related to the iron metabolism was differentially upregulated following the *dtxR* overexpression [54]. In addition, the bioinformatic predictions and DNA band-shift analyses indicated that at least 64 genes encoding multiple physiological functions, such as the iron transport and utilization, central carbohydrate metabolism, and transcriptional regulation, were directly controlled by the DtxR proteins [88]. In conclusion, the DtxR protein is important for maintaining the intracellular iron homeostasis, and the heme *b* production can possibly be further improved by fully elucidating its regulatory mechanism in the future.

**Table 1 molecules-28-03633-t001:** Recent advances in the biosynthesis of heme *b*.

Chassis	5-ALA Synthetic Pathway	Heme *b* Synthesis Pathway	Detection	Description	Culture Mode	Titer (mg/L)	Heme Secretion Ratio (%)	Ref.
*E. coli* K-12 JM109	C4	PPD	LC-MS	Overexpression of *alaS_RC_*, *pbgS*, *hmbS*, *uroS*, *uroD_SP_*, *cgdC*, and *ppfC_BS_*	Flask	2.03 ± 0.18	ND ^1^	[10]
*E. coli* W3110	C4	-	Spectrophotometer	Overexpression of *alaS_RS_*, *macB*, and *dctA*	Fed-batch	6.4	ND ^1^	[60]
*E. coli* W3110	C4	-	LC-MS	Overexpression of *alaS_RS_*, *macB*, and *dctA* and optimization of the fermentation conditions	Fed-batch	0.12	ND ^1^	[90]
*E. coli* W3110 (DE3)	C4	-	HPLC	Overexpression of *alaS_RS_*, *macB*, and *coaA* and the addition of glycine, succinate	Flask	0.49 ^2^	ND ^1^	[65]
*E. coli* W3110	C4	-	LC-MS	Overexpression of *alaS_RS_*, *coaA* and the addition of glycine, succinate, and FeCl_3_·6H_2_O	Flask	9.1 ^2^	ND ^1^	[64]
*E. coli* BL21 (DE3)	C5	PPD	HPLC	Overexpression of *gluTR^fbr^*, *gsaM*, *pbgS*, *hmbS*, *uroS*, *uroD*, *cgdC*, *pgoX*, and *ppfC*, and deletion of *yfeX*, *ldhA*, and *pta*	Flask	7.88	16.0	[9]
*E. coli* BL21 (DE3)	C5	PPD	HPLC	Overexpression of *gluTR^fbr^*, *gsaM*, *pbgS*, *hmbS*, *uroS*, *uroD*, *cgdC*, *pgoX*, *ppfC*, and *ccmABC*, deletion of *yfeX*, *ldhA*, and *pta* and optimization of the fermentation conditions	Fed-batch	239.2	63.3	[9]
*E. coli* BL21 (DE3)	C5	PPD	HPLC	Overexpression of *gluTR^fbr^*, *gsaM*, *pbgS*, *hmbS*, *uroS*, *uroD*, *cgdC*, *pgoX*, *ppfC*, and *ccmABC*, deletion of *yfeX*, *ldhA*, and *pta* and optimization of the fermentation conditions	Fed-batch	1034.3	45.5	[12]
*C. glutamicum* ATCC 13032	C5	-	Fluorescence	Overexpression of *gluTS_EC_*, *gluTR_EC_*, and *gsaM_SA_*	Flask	4.22 ± 0.62	ND ^1^	[13]
*C. glutamicum* ATCC 13826	C5	CPD	HPLC	Overexpression of *gluTR_ST_*, *gsaM_EC_*, and *chdC^ATG^*	Flask	27.22 ± 0.65	ND ^1^	[55]
*C. glutamicum* ATCC 14067	C4+C5	CPD	HPLC	Overexpression of *gluTR^M^_ST_*, *gsaM_EC_*, *alaS_RC_*, *uroD^AUG^*, *cgoX*, *cpfC*, *chdC*, *dtxR*, *hrtA*, and *hrtB*, and deletion of *hrrS*, *htaA*, and *hmuT*	Flask	38.16 ± 0.52	21.7	[54]
*C. glutamicum* ATCC 14067	C4+C5	CPD	HPLC	Overexpression of *gluTR^M^_ST_*, *gsaM_EC_*, *alaS_RC_*, *uroD^AUG^*, *cgoX*, *cpfC*, *chdC*, *dtxR*, *hrtA*, and *hrtB*, deletion of *hrrS*, *htaA*, and *hmuT*, optimization of the fermentation conditions, and addition of the cell wall inhibitor ethambutol	Fed-batch	309.18 ± 16.43	78.58	[54]
*C. glutamicum* ATCC 14067	C4+C5	CPD	HPLC	Overexpression of *gluTR^M^_ST_*, *gsaM_EC_*, *alaS_RC_*, *uroD^AUG^*, *cgoX*, *cpfC*, *chdC*, *dtxR*, *hrtA*, and *hrtB*, deletion of *hrrS*, *htaA*, and *hmuT*, and optimization of the fermentation conditions	Fed-batch	111.87 ± 6.48	91.25	[54]
*S. cerevisiae* BY4741	C4	PPD	Fluorescence	Overexpression of *alaS*, *pbgS*, *hmbS*, *cgdC*, *pgoX*, *ppfC*, and *fet4* and deletion of *shm1*, *hmx1*, *gcv2*, and *gcv1*	Flask	53.5	ND ^1^	[14]
*S. cerevisiae* CEN.PK2-1C	C4	PPD	Fluorescence	Overexpression of *alaS* (copy number is 2), *cgdCP*, *pgoXP40-539*, and *ppfCP31-393* and deletion of *hmx1*	Flask	0.3	ND ^1^	[67]
*Pichia pastoris* L10A1T	C4	PPD	Heme detection kit	AOX1p-HmbS-AOX1p-AlaS-AOX1p-PbgS-AOX1p-UroD-AOX1p-PgoX-AOX1p-CgdC-AOX1p-UroS-AOX1p-PpfC integrated into the his4 locus and complementation of *ku70*	Fed-batch	132	ND ^1^	[91]

^1^ ND: not determined. ^2^ The units are μmol/g-DCW. The actual heme titers could not be determined because the biomass concentrations were not reported.

## 4. Detection of Heme *b*

The detection of heme *b* is important for the study of the synthetic pathways, metabolic mechanisms, and practical applications. The measurement methods that have been developed can be divided into the direct and indirect detection of heme *b* (Table 2).

### 4.1. Spectrophotometric Heme b Assay of Pyridine Hemochrome

When the nitrogen ligands in the protein-bound heme *b* were replaced by pyridine in alkali, it was converted into hemochrome, which can be detected spectrophotometrically [92]. The quantification of the hemochrome can be conducted by contrasting the different spectra of the reduced and oxidized compound [93]. First, the hemochrome sample was divided equally into two parts. The first part was reduced with an appropriate amount of sodium sulfite and the absorbance was measured at 557 nm. To ensure that the reaction proceeded completely, a small amount of sodium sulfite was further added to the tube and the spectra was scanned again until there was no difference. Potassium ferricyanide was added to another tube to oxidize the hemochrome and its absorbance was measured at 541 nm. Finally, the heme *b* concentration was calculated from the difference in the absorption between the peak at 557 nm and the trough at 541 nm [94]. It is worth noting that the specific measurement wavelength should be selected after obtaining a baseline absorption spectrum in the range of 500 to 600 nm due to the possible differences between the instruments. This method can also be used to simultaneously determine heme *a*, *b*, and *c*, but the sensitivity to error is greater than when measuring a pure compound at a single wavelength [95]. This method is simple to operate but is sensitive to interference from the impurities with the same spectral characteristics.

### 4.2. Heme b Assay Based on Protoporphyrin Fluorescence

In this assay, heme *b* was heated in a strong oxalic acid solution to produce protoporphyrin, which can be quantified due to its unique fluorescence properties [96]. When the samples already contained significant amounts of porphyrins or other substances that fluoresced at the wavelength of determination, they needed to be accurately subtracted using a reagent blank. After adding a high concentration of oxalic acid to the samples, they were divided equally into two parts, one of which was used as the measurement sample and the other as the blank. The high concentrations of oxalic acid close to the saturation usually required heating to dissolve. The measurement sample was then heated in a heating block set at 100 °C for 30 min, while the blank sample was stored at room temperature. Finally, the fluorescence of the porphyrin was measured in both samples using an excitation wavelength of 400 nm and an emission wavelength of 662 or 608 nm. The 662 nm emission peak had a lower fluorescence than the 608 nm peak, but there was also less scattering and other interference [93]. It should be noted that the actual peak may vary from instrument to instrument. The difference in the fluorescence between the two groups was used to calculate the heme *b* concentration. However, the high concentration of oxalic acid used in this method was supersaturated at room temperature and needed to be heated to dissolve, which required a high degree of operational rigor.

### 4.3. Heme b Analysis via High-Performance Liquid Chromatography (HPLC)

The HPLC separation of each chemical entity from the sample mixture is based on its different affinity for the adsorbent material in the column or mobile phase, allowing the various components to move and separate at different rates. The detection efficiency of the method depends mainly on some intrinsic tunable parameters of the mobile phase, such as the composition, flow rate and pH; the type and nature of the stationary phase; and the environmental factors. HPLC is relatively complex, but it is more accurate and can also detect the intermediate products in the heme *b* pathway. It is, therefore, the most commonly used assay for the metabolic engineering of the heme *b* production today. The addition of an acetone hydrochloric acid solution to the sample dissolved the heme *b* molecules in acetone and the multiple extractions allowed for a full dissolution of heme *b*. The resulting mixture was then separated on a C18 HPLC column [97], and the detection wavelength for heme *b* was generally 400 nm. The specific buffer composition and elution program needed to be optimized for each HPLC system and column. Therefore, different researchers developed different protocols [9,10,12,54,75,90]. For more accurate identification about the intermediate metabolites, liquid chromatography-mass spectrometry (LC-MS) is also commonly used [10,90].

### 4.4. Heme b Biosensors

The metabolic regulation is a dynamic and real-time monitoring process of intracellular heme *b* that is necessary for many applications. This is supported by the design of the heme *b* biosensors, which can be used to assess the dynamic changes of the heme *b* concentrations. Hana et al. [98] constructed the genetically encoded ratiometric fluorescent heme sensors, HS1, in the unicellular eukaryote *S. cerevisiae*. The HS1 contains a heme-binding domain, the His/Met coordinating 4-alpha-helix bundle hemoprotein cytochrome b_562_ (Cyt b_562_), which binds to the fluorescent proteins (EGFP) and Katushka 2 (mKATE2). *Holo*-Cyt b_562_ is a fluorescence resonance energy transfer (FRET) receptor for the EGFP excitation. Subsequently, heme-LBB is used as a fusion structure consisting of a green fluorescent protein (GFP) and hemoglobin (Hb) to reflect the intracellular heme *b* levels [14]. Heme *b* is co-translationally incorporated into the hemoglobin part of the biosensor polypeptide and facilitates its correct folding. The GFP-Hb fusion that is bound to heme *b* is active and fluorescent, while the empty biosensor molecule folds incorrectly and is susceptible to degradation. As a consequence, an increase in the heme *b* production is correlated closely with the increase in the fluorescence of heme-LBB. The heme *b* biosensor enables the conversion of the heme concentration into the GFP fluorescence intensity and does not require an additional extraction process for the measurement. This method can be used to quantify the intracellular heme *b* concentration in real time, but it is not applicable to the extracellular samples.

**Table 2 molecules-28-03633-t002:** Detection methods of heme *b*.

Method Name	Sample	Description of the Method Conditions	Detection Time (min)	Linearity Range, Limits of Detection, and Quantification (LOQ)	Ref.
Spectrophotometric heme *b* assay of pyridine hemochrome	Tissue homogenate, homogenate or sonicated lysate of the tissue culture cells	Add 1 mL of fermentation broth to the appropriate amount of the pyridine-NaOH solution and sulfite sodium. Mix well and leave for 30 min. The samples were Separate the samples into two cuvettes and measure the absorbance at 557 nm. Add 0.01 mL of 3 M potassium ferricyanide to one cuvette and measure the absorbance at 541 nm. Calculate the difference in the absorbance values.	30	<7 μg/mL	[92]
Heme *b* assay based on protoporphyrin fluorescence	Tissue culture cells	Collect tissue culture cells, add 0.5 mL of a 2 M oxalic acid solution and mix thoroughly. Put the sample tubes in a heating block set at 100 °C for 30 min. Do not heat the blanks. Read the fluorescence of porphyrin using an excitation wavelength of 400 nm and an emission wavelength of 662 or 608 nm. Subtract the blank values (parallel unheated samples in oxalic acid).	30	1 nM~1 μM	[93]
Heme *b* analysis via high-performance liquid chromatography (HPLC)	Tissue culture cells or tissue (sonicated lysate or homogenate)	Harvest 1 mL of cell pellet using centrifugation (13,000 rpm, 4 °C, 3 min). Add 1 mL of an acetone: HCl (95:5) solution to the cell collection tube. Vortex the mixture and dilute with 1 mL of 1 M NaOH. Filter the disrupted intracellular samples and supernatants using mixed cellulose ester (MCE) filters. Separate the samples on a C18-HPLC column using a linear gradient method of 20–95% of solvent A in B at 40 °C. Solvent A is a 10:90 (*v*/*v*) mixture of HPLC grade methanol and acetonitrile, and solvent B is 0.5% (*v*/*v*) trifluoroacetic acid (TFA) in HPLC grade water. The flow rate was 1 mL/min for 40 min, and the elution was followed by recording the absorbance at 400 nm.	40	ND ^1^	[54]
Heme *b* biosensors	Tissue culture cells	Express heme-LBB using the copper-inducible promoter CuPi of *S. cerevisiae*. Place 200 µL of the fermentation broth in a black 96-well plate and measure the fluorescence of the GFP at an excitation wavelength of 488 nm and an emission wavelength of 507 nm.	ND ^1^	ND ^1^	[14]

^1^ ND: Not determined.

## 5. Concluding Remarks and Future Perspectives

As the most common porphyrin, heme *b* has significant potential for use in healthcare, food processing, and chemical production. However, the current chemical synthesis and bio-extraction techniques are not sufficient to sustain the market demand. As a consequence, the microbial cell factories for the heme *b* production have drawn increasing attention over the past two decades. To date, the highest heme *b* titer for microbial fermentation has reached 1034.3 mg/L [12]. However, there is still a long way to go before heme *b* can be biosynthesized on an industrial scale, and some common problems still need to be addressed. (I) Only two model organisms have been used for the biosynthesis of heme *b*, and the exploration of the natural high producers as chassis strains is inadequate. (II) The regulatory mechanism of the heme *b* synthesis is highly complex and not yet fully understood. (III) Screening the metabolic engineering target is mainly limited to the heme *b* biosynthesis pathway genes. (IV) The long heme *b* synthesis pathway imposes a certain metabolic burden on a single-cell system.

### 5.1. Exploration of Natural High-Producers as Novel Chassis Strains

The analysis of 982 representative prokaryotic genomes showed that 806 organisms (82.1% of the total) were capable of a *de novo* synthesis of UPG III, 713 of which (72.6% of the total) had the ability to synthesize heme *b* from 5-ALA [11]. However, the strains currently used in the microbial synthesis of heme *b* for research include only two model species, *E. coli* and *C. glutamicum*. Obviously, many of the natural high-producing strains have not been exploited for the heme *b* biosynthesis. It can be predicted that in the future, a full screening of the natural hosts with innate advantages in the heme *b* biosynthesis could lead to a high-yield production of heme *b*. For example, *S. cerevisiae* [67] and *Pichia pastoris* [91] play important roles in the microbial production of hemoglobin. However, they do not require expensive media and inhibitors, such as ethambutol, to minimize the binding of heme *b* to the cell membranes. The analysis of the genetic information on the heme *b* biosynthesis currently available in research databases allows for a better screening of the natural high producers, and the process can be accelerated by developing a high-quality standardized database. Czajka et al. [99] integrated knowledge mining, feature extraction, genome-scale modeling (GSM), and machine learning [83] to develop a model for predicting the potential titers for the different chemicals produced in *Yarrowia lipolytica*. Thus, by using species-specific GSMs in combination with machine learning, the range of the heme *b*-producing strains can be further extended.

### 5.2. Elucidation of Heme-Regulating Mechanisms

Three biosynthetic pathways of heme *b* are now widely recognized, but the regulatory mechanisms are not well understood. Understanding the regulatory properties of the key enzymes of the synthetic pathway, such as PpfC and CpfC, is an important part for achieving their precise regulation. PpfC and CpfC are responsible for a key step in the heme *b* synthesis, the insertion of the ferrous ions into the porphyrin ring. The regulation of the key enzymes will directly affect the performance of the artificial cell factory. Based on recent advances in artificial intelligence (AI), the AlphaFold2 software has been developed to predict the 3D shape of proteins based on their genetic sequences with, for the most part, pinpoint accuracy [100]. Once the spatial structure of PpfC or CpfC is obtained, molecular docking can be used to determine the binding mode of the ferrous ions. New predictive tools can be integrated with multiple platforms to produce models of the protein complexes by designing new neural network AI. In addition, systems biology has enabled a shift from the traditional single-omics level to a multi-omics integrated analysis model for large-scale studies of whole cell systems. Multi-omics integrated analysis machine learning methods and algorithms, such as AMARETTO [101], can provide a comprehensive and deep understanding of the regulatory relationships between the various biomolecules, and then resolve the genetic regulatory mechanisms of the organisms. The rational design of the enzymes and the fine regulation of the metabolic pathways can be achieved by fully understanding the regulatory mechanism of the synthetic pathway to promote the efficient synthesis of heme *b*.

### 5.3. Target Gene Mining and Genome-Scale Design

Currently, the engineering targets are limited to the genes encoding the enzymes that are directly responsible for the heme *b* synthesis. However, genome-scale target identification is expected to achieve a higher heme *b* production by subtly affecting the other nodes in the metabolic network. The genome-scale metabolic modeling (GEM) allows for the expression, transcription, and translation of all the host genes to be taken into account, fully simulating the metabolic changes during growth and identifying the key targets for the heme *b* biosynthesis through a metabolic flux analysis. Ishchuk et al. [14] identified 84 target genes for balancing the biomass and increasing the heme *b* production by using the *S. cerevisiae* Yeast8 model. To further optimize the phenotypic prediction of the metabolism, ecYeast8 was applied to the enzyme usage variability analysis and mechanistic simulations for the individual gene modifications for the heme *b* production, which ultimately resulted in a 70-fold higher intracellular heme *b* accumulation. The metabolic models with multiple constraints integrating the multi-omics information can lead to more accurate results in pathway analyses and phenotype predictions. For example, the *E. coli* metabolic models with enzymatic and thermodynamic constraints (ETGEMs) resulted in more realistic production pathways and L-arginine yields [102]. The synthesis of heme *b* is subject to complex regulatory mechanisms, and their appropriate integration into the process of solving metabolic network models can better guide the practice of metabolic engineering. It is foreseeable that genome-scale target mining will be much faster than the characterization and functional elucidation of the heme *b* biosynthetic pathway enzymes.

### 5.4. Construction of Artificial Microbial Consortia for the Heme b Synthesis

Although it is a natural product with a long and complex synthesis pathway, heme *b* has the natural advantage of being biosynthesized by many potential microbial cell factories. Inevitably, the overexpression of the enzymes in the heme *b* synthesis pathway on a single-cell system can lead to an excessive accumulation of intermediate metabolites that not only cause carbon loss, but also have a toxic effect on the cells. The dynamic regulatory systems, in response to the intermediate metabolites, could couple growth to production, thus alleviating the metabolic burden. On the other hand, MMME was used to regulate the expression of the genes in the heme *b* synthetic pathway, but the instability of the plasmids required the development of more powerful chromosomal integration techniques or strategies, such as the CRISPR–dCas12a-mediated multiple pathway optimization framework [103]. In addition, a novel alternative strategy to decrease the metabolic burden of single-cell systems is the construction of artificial microbial consortia. Artificial microbial consortia can separate the biosynthetic modules into different microorganisms to achieve an efficient synthesis of the complex chemicals by applying a ‘divide-and-conquer’ strategy. UPG III is the core structure of tetrapyrroles, but when UroS is not present or has a very low activity, HMB will spontaneously generate the by-product UPG I [11]. Studies have shown that UPG III is the most abundant isomer formed by PBG under acidic conditions [26]. Thus, a modular co-culture system can be formed by placing the UPG III synthesis module in an acid-tolerant strain lacking the ability to synthesize heme *b* (or containing an incomplete pathway) and placing the heme *b* synthesis module in strains that are tolerant to high concentrations of heme *b*. These new synthetic strategies not only reduce the metabolic burden of the heme *b* synthesis in single strains, but also allow for the flexible optimization of the individual modules. The successful construction of artificial microbial consortia often relies on dynamic adjustments between the organisms, so the construction of a UPG III biosensor and a heme *b* biosensor (e.g., based on riboswitches [72,104]) is a key step to further improve the heme *b* titers.

## Figures and Tables

**Figure 1 molecules-28-03633-f001:**
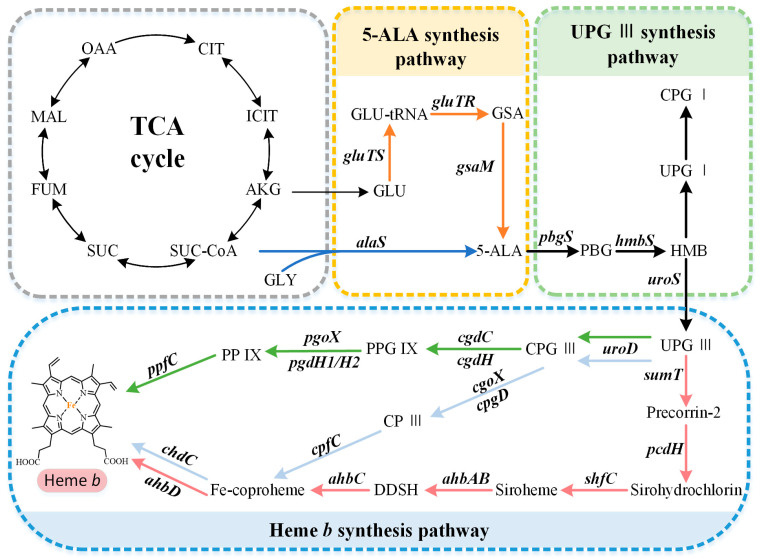
Heme *b* biosynthesis pathway. The overall biosynthetic pathway of heme *b* can be divided into three parts. The orange arrows indicate the C5 pathway, the blue arrows indicate the C4 pathway, the green arrows indicate the protoporphyrin-dependent (PPD) pathway, the light blue arrows indicate the coproporphyrin-dependent (CPD) pathway, and the red arrows indicate the siroheme-dependent (SHD) pathway. OAA, oxaloacetate; MAL, malate; FUM, fumarate; SUC, succinate; SUC-COA, succinyl-CoA; α-KG, α-oxoglutarate; GLY, glycine; GLU, glutamate; GLU-tRNA, glutamyl-tRNA; GSA, glutamate-1-semialdehyde; 5-ALA, 5-aminolevulinic acid; PBG, porphobilinogen; HMB, hydroxymethylbilane; UPG I, uroporphyrinogen I; CPG I, coproporphyrinogen I; UPG III, uroporphyrinogen III; CPG III, coproporphyrinogen III; PPG IX, protoporphyrinogen IX; PP IX, protoporphyrin IX; CP III, coproporphyrin III; DDSH, 12,18-didecarboxysiroheme.

**Figure 2 molecules-28-03633-f002:**
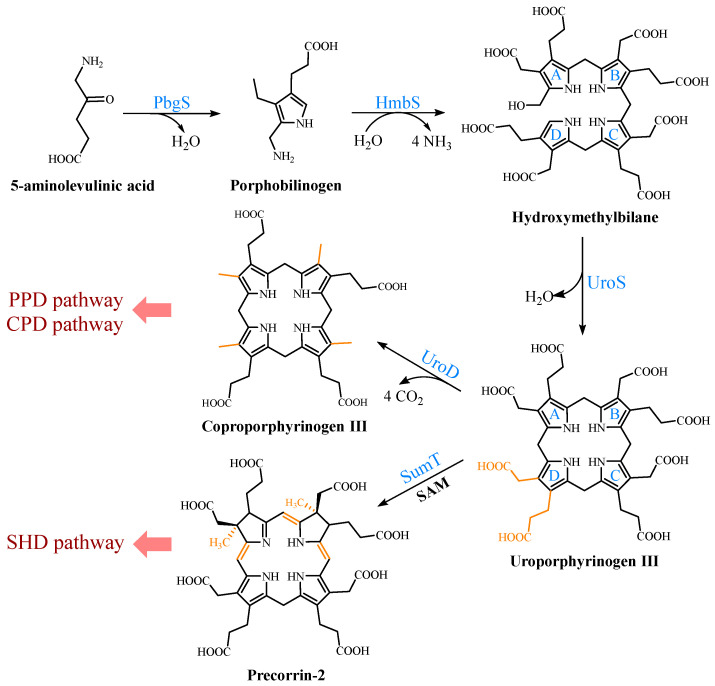
Formation of uroporphyrinogen III (UPG III) and the branching of the heme *b* biosynthesis. In nature, the pathway from 5-aminolevulinic acid (5-ALA) to UPG III is a highly conserved three-enzyme cascade. UPG III is decarboxylated by uroporphyrinogen III decarboxylase to form coproporphyrinogen III (CPG III), a reaction shared by the protoporphyrin-dependent (PPD) and coproporphyrin-dependent (CPD) pathways. In the siroheme-dependent (SHD) pathway, the C2 and C7 positions of UPG III are methylated by the SAM-dependent uroporphyrinogen III methyltransferase (SumT) to generate precorrin-2.

**Figure 3 molecules-28-03633-f003:**
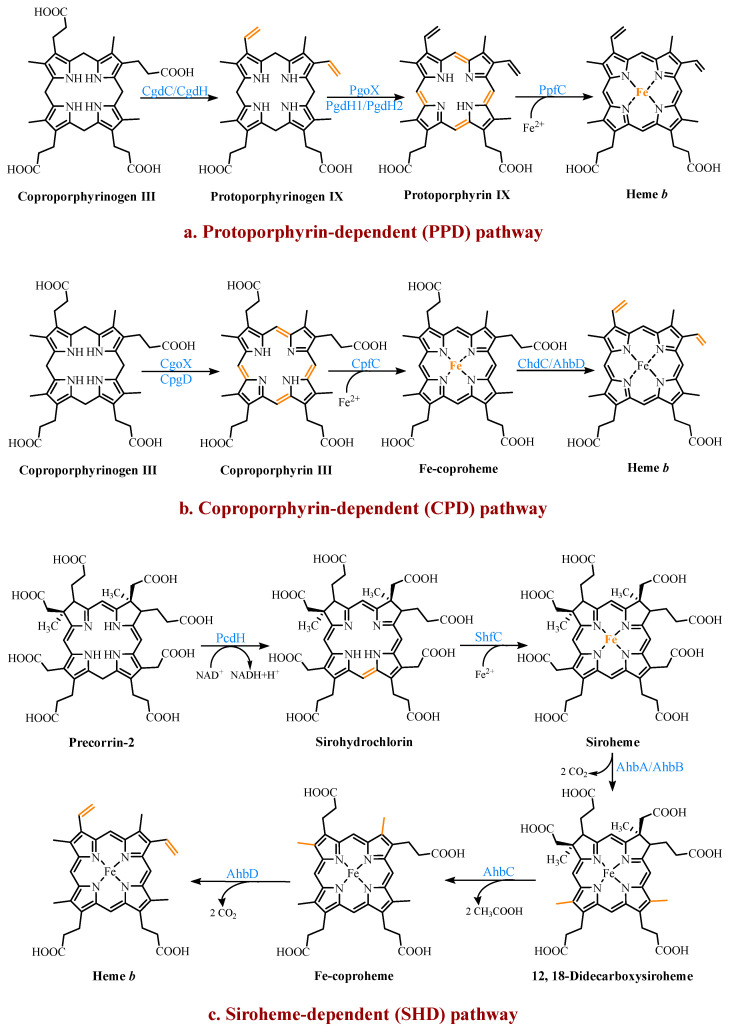
The three known pathways for the heme *b* biosynthesis. In the protoporphyrin-dependent (PPD) and coproporphyrin-dependent (CPD) pathways, the four methyl groups of heme *b* are the result of the stepwise decarboxylation of the acetate side chains of uroporphyrinogen III (UPG III), with four methyl groups produced at the C2, C7, C12, and C18 positions. The SHD pathway involves the S-adenosyl-L-methionine (SAM)-dependent methylation of the C2 and C7 positions of the tetrapyrrole backbone.

**Figure 4 molecules-28-03633-f004:**
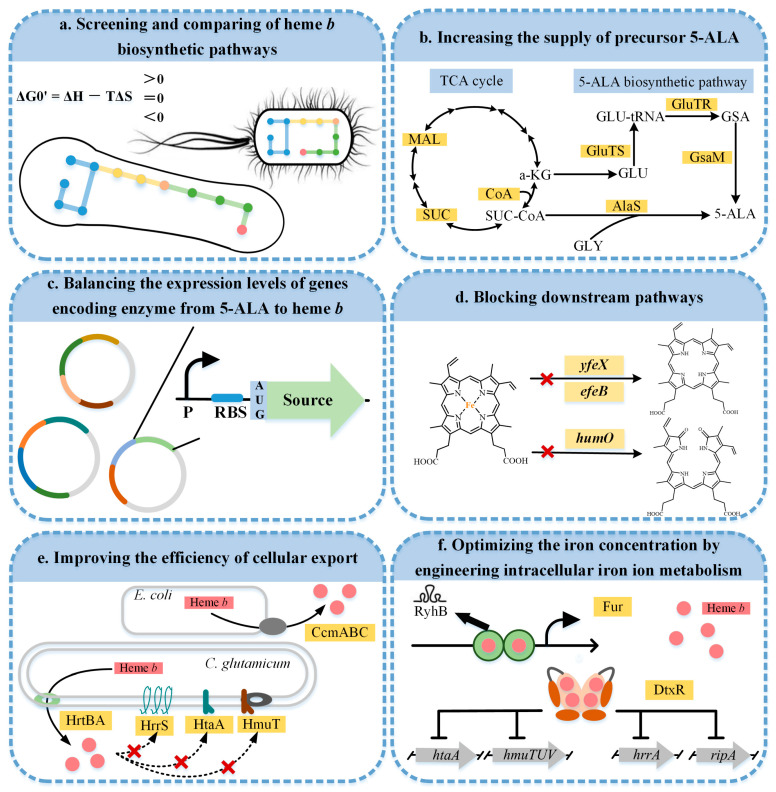
Metabolic engineering strategies for the heme *b* biosynthesis. A red cross indicates that the corresponding pathway is blocked. MAL, malate; SUC, succinate; SUC-COA, succinyl-CoA; CoA, coenzyme A; α-KG, α-oxoglutarate; GLY, glycine; GLU, glutamate; GLU-tRNA, glutamyl-tRNA; GSA, glutamate-1-semialdehyde; 5-ALA, 5-aminolevulinic acid; GluTS, glutamyl-tRNA synthetase; GluTR, glutamyl-tRNA reductase; GsaM, glutamate-1-semialdehyde-2,1-aminomutase; AlaS, 5-aminolevulinic acid synthase; HmuO, heme oxygenase; Ccm ABC, cytochrome c export protein; HrtBA, heme regulated transporter A and B; Fur, ferric uptake regulator; DtxR, diphtheria toxin repressor.

## Data Availability

No new data were created.

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
