# Peer review of "Microbial Synthesis of Heme b: Biosynthetic Pathways, Current Strategies, Detection, and Future Prospects"

_molecules, 2023, doi:10.3390/molecules28083633_

Round 1
Reviewer 1 Report
The review provides the first systematic summary of the progress in the microbial synthesis of heme b. I find the instruction of the paper quite good and logical ideas. However, the comparison of metabolic engineering strategies for heme b biosynthesis should be highlighted in the text. Moreover, the author should consider writing the scientific names of species microbes and genes (italic format).
Listed are some comments regarding the submitted manuscript:
1. Line 18: what is “new technologies”?
2. Concluding Remarks and Future Perspectives: please provide the future approaches to decrease the metabolic burden on a single-cell system for hem b biosynthesis.
Reviewer 2 Report
The manuscript is well written, is easy to read and is properly presented. The paper discuses the progress in microbial synthesis of heme b. Analyze the pathways for the biosynthesis of heme b and discusses the methods used to quantify heme b. A deep English review must be performed, some examples:
Line 60: “biosyntheiss” must be “biosynthesis”
Line 267: “screening and screening and comparing” must be “Screening and comparing”
Line 523: “seme b” must be “heme b”
Author Response
Dear editors and reviewers,
Thank you for your advice and comments concerning our manuscript entitled “Microbial Synthesis of Heme b: Biosynthetic Pathways, Current Strategies, Detection and Future Prospects” (molecules-2281374). The comments were highly valuable and very helpful for improving our paper. The manuscript has been revised and submitted with new lines and page numbers. The revised parts are shown in red. We appreciate the editors’ and reviewers’ comments, and hope that the corrections will meet with approval. The corrections and responses to the reviewer’s comments are listed below.
Responses to reviewer #2:
The manuscript is well written, is easy to read and is properly presented. The paper discuses the progress in microbial synthesis of heme b. Analyze the pathways for the biosynthesis of heme b and discusses the methods used to quantify heme b.
Response: Thank you for your comments on our manuscript.
Point: A deep English review must be performed, some examples:
Line 60: “biosyntheiss” must be “biosynthesis”
Line 267: “screening and screening and comparing” must be “Screening and comparing”
Line 523: “seme b” must be “heme b”
Response : Thank you for your suggestion. According to your suggestion, we checked the full article and revised it. Here, only few examples were listed:
Line 65: “Heme b biosyntheiss pathway” was revised as “Heme b biosynthesis pathway”.
Line 66-67: ”The orange arrow indicates the C5 pathway, the blue arrow indicates the C4 biosynthesis pathway” was revised as “The orange arrows indicate the C5 pathway, the blue arrows indicate the C4 pathway”.
Line 148-149:”the decarboxylation of UPG III catalyzed by uroporphyrinogen III decarboxylase [27] forms coproporphyrinogen III (CPG III)” was revised as ”the decarboxylation of UPG III catalyzed by uroporphyrinogen III decarboxylase (UroD) [27] forms coproporphyrinogen III (CPG III)”.
Line 284: “Screening and screening and comparing of heme b biosynthetic pathways” was revised as “Screening and comparing of heme b biosynthetic pathways”.
Line 547: “In this assay, seme b is heated in a strong oxalic acid solution to produce protoporphyrin, which can be quantified due to its unique fluorescence properties [95].” was revised as “In this assay, heme b is heated in a strong oxalic acid solution to produce protoporphyrin, which can be quantified due to its unique fluorescence properties [95].”
Reviewer 3 Report
This manuscript is well structured and the language flows without errors. The content is detailed and well argued. It can be published after some refinement.
1. Examples of other ways of extracting heme b that are not environmentally friendly.
2. What specific disadvantages can be optimized for microbial production of heme b compared with other methods?
3. At present, there are more detection methods, but what is the principle of determining heme b by HPLC and biosensor, and why can promote the production of heme b by microorganisms?
4. Why is the oldest SHD suitable for microbial production of heme b, and which specific regulatory mechanisms indicate that it is more suitable.
5. The final conclusion and future outlook, future outlook can add more content.
